# The RNA-Binding Protein ELAVL1 Regulates Hepatitis B Virus Replication and Growth of Hepatocellular Carcinoma Cells

**DOI:** 10.3390/ijms23147878

**Published:** 2022-07-17

**Authors:** Hiroaki Kanzaki, Tetsuhiro Chiba, Tatsuya Kaneko, Junjie Ao, Motoyasu Kan, Ryosuke Muroyama, Shingo Nakamoto, Tatsuo Kanda, Hitoshi Maruyama, Jun Kato, Yoh Zen, Ai Kotani, Kazuma Sekiba, Motoyuki Otsuka, Masayuki Ohtsuka, Naoya Kato

**Affiliations:** 1Department of Gastroenterology, Graduate School of Medicine, Chiba University, 1-8-1 Inohana, Chuo-ku, Chiba 260-8670, Japan; kanzakih@chiba-u.jp (H.K.); kanekota@chiba-u.jp (T.K.); aojunjie@chiba-u.jp (J.A.); kan.motoyasu@chiba-u.jp (M.K.); nakamotoer@faculty.chiba-u.jp (S.N.); kato.jun@chiba-u.jp (J.K.); kato.naoya@chiba-u.jp (N.K.); 2Department of Molecular Virology, Graduate School of Medicine, Chiba University, 1-8-1 Inohana, Chuo-ku, Chiba 260-8670, Japan; muroyama-dm@chiba-u.jp; 3Department of Gastroenterology and Hepatology, Nihon University School of Medicine, 30-1 Oyaguchi-Kamicho, Itabashi-ku, Tokyo 173-8610, Japan; kanda2t@yahoo.co.jp; 4Department of Gastroenterology, Graduate School of Medicine, Juntendo University, 2-1-1 Hongo, Bunkyo-ku, Tokyo 113-8421, Japan; maru-cib@umin.ac.jp; 5Institute of Liver Studies, King’s College Hospital, London SE5 9RS, UK; yoh.1.zen@kcl.ac.uk; 6Division of Hematological Malignancy, Institute of Medical Sciences, Tokai University, 143 Shimokasuya, Isehara, Kanagawa 259-1193, Japan; aikotani@k-lab.jp; 7Department of Gastroenterology, Graduate School of Medicine, The University of Tokyo, 7-3-1 Hongo, Bunkyo-ku, Tokyo 113-8655, Japan; skb73.0@gmail.com (K.S.); otsukamoto@gmail.com (M.O.); 8Department of General Surgery, Graduate School of Medicine, Chiba University, 1-8-1 Inohana, Chuo-ku, Chiba 260-8670, Japan; otsuka-m@faculty.chiba-u.jp

**Keywords:** ELAVL1, HBV, HBx, HCC, RNA-binding protein

## Abstract

Previous RNA immunoprecipitation followed by proteomic approaches successfully demonstrated that Embryonic Lethal, Abnormal Vision, Drosophila-Like 1 (ELAVL1) interacts with hepatitis B virus (HBV)-derived RNAs. Although ELAVL family proteins stabilize AU-rich element (ARE)-containing mRNAs, their role in HBV transcription remains unclear. This study conducted loss-of-function assays of ELAVL1 for inducible HBV-replicating HepAD38 cells and *HBx*-overexpressed HepG2 cells. In addition, clinicopathological analyses in primary hepatocellular carcinoma (HCC) surgical samples were also conducted. Lentivirus-mediated short hairpin RNA knockdown of *ELAVL1* resulted in a decrease in both viral RNA transcription and production of viral proteins, including HBs and HBx, probably due to RNA stabilization by ELAVL1. Cell growth of HepAD38 cells was more significantly impaired in *ELAVL1*-knockdown than those in the control group, with or without HBV replication, indicating that ELAVL1 is involved in proliferation by factors other than HBV-derived RNAs. Immunohistochemical analyses of 77 paired HCC surgical specimens demonstrated that diffuse ELAVL1 expression was detected more frequently in HCC tissues (61.0%) than in non-tumor tissues (27.3%). In addition, the abundant expression of ELAVL1 tended to affect postoperative recurrence in HBV-related HCC patients. In conclusion, ELAVL1 contributes not only to HBV replication but also to HCC cell growth. It may be a potent therapeutic target for HBV-related HCC treatment.

## 1. Introduction

Hepatitis B virus (HBV) infection is one of the global health concerns. Despite the existence of a prophylactic vaccine, approximately 240 million individuals worldwide are currently infected and at a high risk of developing cirrhosis and hepatocellular carcinoma (HCC) [1,2]. HBV virions contain partially double-stranded, relaxed circular DNA (3.2 kb genome in length), from which covalently closed circular DNA (cccDNA) is formed. HBV cccDNA is persistently present in the nucleus of hepatocytes in an episomal state and serves as a template for viral transcription [3,4]. It is known that five types of RNA are transcribed using cccDNA as a template [5]. There are two kinds of 3.5 kb mRNA, namely precore mRNA and pregenomic RNA. Precore mRNA translates HBeAg, and pregenomic RNA translates HBc and polymerase. HBs antigens (large, middle, and small S proteins), which are envelope proteins, are translated from 2.4/2.1 kb mRNA. The 0.7 kb mRNA encodes an HBx protein that regulates transcription of viral mRNA from cccDNA. Previously, it was believed that HBV-derived RNAs were only templates for viral protein translation and intermediates for the viral DNA synthesis by reverse transcription for replication, but HBV-derived RNAs have recently been found to also have crucial pathological functions [6,7,8].

The currently available HBV therapeutics, including nucleos(t)ide analogs (NAs), efficiently suppress viral reverse transcription and thereby reduce serum levels of HBV-DNA. However, episomal viral cccDNA, as well as transcripts and proteins from cccDNA, remain expressed during these treatments [9,10]. In this context, both transcripts and proteins from cccDNA might be a potential new therapeutic target. Sekiba et al. performed a comprehensive RNA precipitation assay to determine host proteins that interact with HBV-derived RNAs including the S, X, and core regions [11]. Consequently, they successfully identified 15 candidate proteins. Among them, the RNA-protein binding has been successfully confirmed by Western blotting for six proteins, including ELAVL1 (also known as HuR). ELAVL1 is a ubiquitously expressed RNA-binding protein in many types of cells and plays an important role in mRNA stability regulation. The molecule binds to a specific single-stranded motif in AU-rich elements (ARE) primarily located in the 3′ untranslated regions of early response genes [12]. However, the role of ELAVL1 in not only HBV replication but also in cell growth remains unknown.

In this study, the effects of ELAVL1 on HBV-derived RNAs and proteins in loss-of-function assays using HepAD38 cells, in which HBV production is under the control of a tetracycline, and *HBx*-overexpressed HepG2 cells, were examined. To unveil the association between ELAVL1 expression and clinicopathological features, paired HCC surgical specimens were subjected to immunohistochemical analyses for ELAVL1.

## 2. Results

### 2.1. Regulation of HBV Replication from the Genome in HepAD38 Cells

HBV replication in HepAD38 cells could be controlled using tetracycline, which inhibits HBV transcription from the genome. As expected, the expressions of both total HBV mRNA and HBV 3.5 kb mRNA were strongly repressed under tetracycline treatment (Student’s *t*-test, *p* < 0.05) (Figure 1A).

### 2.2. Stable Knockdown of ELAVL1 in HCC Cells

Loss-of-function assay of ELAVL1 was first conducted using two different shRNA targeting *ELAVL1* (sh-*ELAVL1*-1 and sh-*ELAVL1*-2). A lentiviral vector-expressed shRNA targeting *luciferase* (*Luc)* was used as a control. Quantitative real-time PCR demonstrated that shRNAs against *ELAVL1* significantly repressed *ELAVL1* mRNA levels in HepAD38 cells in the absence of tetracycline (Student’s *t*-test, *p* < 0.05) (Figure 1B). Consistent with these findings, Western blotting also demonstrated decreased ELAVL1 protein levels in the *ELAVL1*-knockdown cells (Figure 1C).

### 2.3. The Effect of ELAVL1 on HBV-Derived RNAs and Proteins

To investigate the effect of ELAVL1 on HBV-derived RNAs, mRNAs extracted from *ELAVL1*-knockdown HepAD38 cells were subjected to quantitative real-time PCR using specific primers for total HBV mRNA and HBV 3.5 kb mRNA (encoding HBV core and pregenomic RNA (pgRNA)). *ELAVL1*-knockdown resulted in a remarkable decrease in not only total HBV mRNA levels but also in HBV 3.5 kb mRNA levels (Student’s *t*-test, *p* < 0.05) (Figure 2A). Next, the role of ELAVL1 on the production of HBs protein was examined. Chemiluminescent enzyme immunoassay (CLEIA) demonstrated that secreted HBsAg levels in the supernatant of *ELAVL1*-knockdown HepAD38 cells were lower than those in the control group (Student’s *t*-test, *p* < 0.05) (Figure 2B). Similarly, quantitative real-time PCR and Western blotting demonstrated that *ELAVL1*-knockdown in the stable *Flag*-*HBx*-overexpressing HepG2 cells obviously reduced *HBx* mRNA (Student’s *t*-test, *p* < 0.05) (Figure 2C) and Flag-HBx protein levels (Figure 2D). Altogether, ELAVL1 plays a significant role in the regulation of not only HBV-derived RNA expression but also in its protein production. To test whether ELAVL1 affects RNA-stability, we conducted time-course RNA-stability assay. RNA was harvested at 0 and 24 h post tetracycline treatment and relative levels of remaining total HBV mRNA and HBV 3.5 kb mRNA were analyzed. The results showed that total HBV mRNA (Figure 2E) and HBV 3.5 kb mRNA (Figure 2F) were destabilized by loss-of-function of ELAVL1 (Student’s *t*-test, *p* < 0.05).

### 2.4. The Role of ELAVL1 in Cell Proliferation Ability

Quantitative real-time PCR demonstrated that shRNAs against *ELAVL1* significantly repressed *ELAVL1* mRNA levels in HepAD38 cells in the presence of tetracycline (Student’s *t*-test, *p* < 0.05) (Figure 3A). Consistent with these findings, Western blotting also demonstrated decreased ELAVL1 protein levels in the *ELAVL1*-knockdown cells (Figure 3B). The alteration of cell proliferation following *ELAVL1*-knockdown was then examined in both the presence and absence of tetracycline. Lentiviral knockdown of *ELAVL1* significantly repressed the cell growth in both tetracycline-treated (Figure 3C) and tetracycline-untreated (Figure 3D) HepAD38 cells (Student’s *t*-test, *p* < 0.05). These results indicate that ELAVL1 regulates not only viral RNA transcription and viral protein production, but also the expression of other tumor-related factors involved in cell proliferation.

### 2.5. Expression of ELAVL1 in Primary HCC Tissues

Next, the expression level of ELAVL1 in HCC tissues was then examined. Analyses of The Cancer Genome Atlas (TCGA) data revealed that there are many cases in which the ELAVL1 mRNA levels were significantly higher in primary HCC than in normal liver (Student’s *t*-test, *p* < 0.05) (Figure 4). To investigate the association between ELAVL1 expression and clinicopathological features, immunohistochemical analyses were conducted on 77 paired samples consisting of tumor and adjacent non-tumor tissues (Figure 5A). ELAVL1 was expressed in the nuclei of hepatocytes at varying frequencies in non-tumor tissues. Of the 77 non-tumor samples, 37 (48.1%) and 21 (27.3%) were classified as partial expression (<50% of nuclei in hepatocytes) and diffuse expression (≥50% of nuclei in hepatocytes) patterns, respectively, but 19 (24.6%) demonstrated no expression of ELAVL1 (Figure 5B). ELAVL1 expression was observed in the nuclei of cholangiocytes but not of infiltrating lymphocytes in the portal area. In tumor tissues, 7 (9.1%), 23 (29.9%), and 47 (61.0%) of the 77 samples were classified as negative, partial expression (<50% of nuclei in HCC cells), and diffuse expression (≥50% of nuclei in HCC cells) patterns, respectively. Among 47 tumor samples with diffuse expression patterns, 13 cases (16.9%) demonstrated ELAVL1 expression not only in the nuclei but also in the cytoplasm of HCC cells. From the perspective of background liver disease, the ELAVL1 positivity rates in HBV-related HCC (93.8%) are higher than those in HCV-related (89.3%) and non-viral HCC (90.9%). However, no significant difference was observed among these groups (Kruskal–Wallis test).

Patient characteristics, including between ELAVL1^low^ HCC and ELAVL1^high^ HCC, were then compared (Table 1). Considering other clinical parameters, including age, gender, fibrosis, AFP levels, BCLC stages, and Edmondson–Steiner grades, no significant differences were also observed between ELAVL1^low^ HCC and ELAVL1^high^ HCC (Mann–Whitney *U* test, Kruskal–Wallis test or Chi-squared test). In addition, no clinical factors that correlated with the ELAVL1-positivity were observed in HBV-related HCC (Mann–Whitney *U* test, Kruskal–Wallis test or Chi-squared test) (Table 2).

Subsequently, prognostic analyses using the Kaplan–Meier method were then conducted according to the ELAVL1 expression. No significant difference in recurrence-free survival (RFS) was observed in all ELAVL1^low^ and ELAVL1^high^ HCC patients (Log-rank test, *p* = 0.08) (Figure 5C). Although no significant difference was found in the RFS between ELAVL1^low^ and ELAVL1^high^ HCC patients, even in those with HBV-related HCC, a trend showing that the RFS in ELAVL1^low^ HCC patients was better than that in ELAVL1^high^ HCC patients was observed (Log-rank test, *p* = 0.06). Notably, no recurrence was observed in patients with ELAVL1^low^ HCC (Figure 5D). Together, it might be that the defective expression of ELAVL1 acts to prevent the recurrence of HBV-related HCC.

## 3. Discussion

NAs, which are widely used for the treatment of chronic HBV infection worldwide, suppress HBV replication by inhibiting the reverse transcription of pregenomic RNA to HBV-DNA [1]. Oral treatment with NAs has a deterrent effect on HCC development through serum HBV-DNA level reduction and liver function improvement [13,14]. In fact, cases of HCC development even when the serum HBV-DNA level is below the detection sensitivity were reported. Although NA has an excellent inhibitory effect on HBV replication, its direct inhibitory effect on cccDNA is limited. The cccDNA in the liver tissue performs a central role as a viral persistent reservoir in HBV infection [15]. Therefore, viral mRNA transcription and viral protein translation using cccDNA as a template usually persist under NA treatment. In fact, the cancer incidence rate has been reported to be higher in HBV patients with high hepatitis B core-related antigen (HBcrAg) levels in the blood, which is said to reflect the amount of cccDNA in the liver tissue, than in patients with low levels [16]. Therefore, cccDNA elimination is the ultimate goal of hepatitis B treatment. Various drug developments have also been attempted. However, clinical application has not been achieved at this time [17,18].

Considering the current status, inhibition of HBV-derived RNA, template for protein synthesis, and viral DNA replication may be an alternative. Generally, RNA is extremely unstable in vivo, but RNA-binding proteins can stabilize RNA by binding to specific RNA sequences and can contribute to a smooth transcriptional regulation [19,20]. Previously, ELAVL1 was successfully demonstrated to bind to HBV-derived RNAs, including the S, X, and core regions [11]. In this study, the role of ELAVL1 in the replication mechanism of HBV was investigated.

A marked suppression of HBV-derived RNA expression was observed in our loss-of-function assay of ELAVL1. Furthermore, the amounts of HBs protein in HepAD38 cells and HBx protein in HepG2 cells were also significantly decreased after *ELAVL1*-knockdown. Given that ELAVL1 contributes to RNA stability by RNA binding [12], HBV-derived RNA expression but also its protein production might be attributable to the stability of HBV-derived RNA. Concordant with this, our data successfully demonstrated that loss-of-function of ELAVL1 resulted in a decrease in RNA stability in HepAD38 cells treated with tetracycline. Taken together, ELAVL1, at least in part, regulates the half-life of HBV-derived RNA. A previous study demonstrated that ELAVL1 interacts with HBV-derived RNAs including the S, X, and core regions using RNA precipitation assay [11]. Future experiments such as pull-down assays using biotinylated transcripts spanning different mRNA regions would be necessary to obtain detailed interaction information between HBV-derived RNA and ELAVL1.

In addition, the present study demonstrated that the proliferative activity of HepAD38 cells was reduced by knockdown of *ELAVL1*, with or without regulation of transcription of HBV-derived RNA by tetracycline. ELAVL1 has been reported to regulate the transcription of growth factors, including c-myc, c-fos, and c-jun, and cyclins, including cyclin A, B1, E, and D1 [21,22]. These findings indicated that ELAVL1 could regulate the cell proliferation ability of HCC cells in HBV replication-independent manners. Altogether, ELAVL1 is a potent therapeutic target for the inhibition of not only HBV replication but also of HCC cell growth.

It is well known that ELAVL1 protein translocates from the cell nuclei (predominantly localized) to the cytoplasm in response to proliferative and stress stimuli, increasing the half-life and/or modulating the translation rate of target mRNAs [23,24]. Concordant with these findings, our immunohistochemical analyses revealed that 70/77 (90.9%) of tumor samples were positive for ELAVL1. The staining strength for ELAVL1 of the nuclei in tumor tissues was stronger than that in adjacent non-tumor tissues in most cases. Although 13 tumor tissues exhibited not only nuclear expression but also cytoplasmic expression of ELAVL1, only two cases demonstrated co-expression of ELAVL1 in the non-tumor tissues. Both an increase in ELAVL1 expression levels and its localization to cytoplasm are reported to be closely associated with malignant transformation [25,26]. The high frequency of simultaneous nucleocytoplasmic expression of ELAVL1 in HCC tissues might suggest that it is closely involved in the post-translational modification of many cancer-related genes.

Previous studies have demonstrated that higher ELAVL1 expression is closely associated with unfavorable prognosis in various cancer types [27,28]. Unexpectedly but importantly, no statistical differences in RFS were observed between ELAVL1^high^ and ELAVL1^low^ HCC patients. When limited to HBV patients, ELAVL1^low^ HCC patients exhibited favorable RFS compared with ELAVL1^high^ patients. These results indicate that ELAVL1 plays a significant role in the development and recurrence of HBV-related HCC. In the present study, cell growth in HepAD38 cells was impaired in the *ELAVL1*-knockdown group independently of HBV replication, but HBx specifically affects the transcription of various cancer-related genes and enhances cell proliferation and capability [29,30]. We showed that ELAVL1 affects HBx, which might be one of the reasons for the results that RFS in ELAVL1^low^ HBV-related HCC patients tended to be better than that in ELAVL1^high^ HBV-related HCC patients. Exploring substantial evidence connecting the role of ELAVL1 in HBV replication and the role of ELAVL1 in HCC is needed to be addressed in future studies through the role of ELAVL1 in real infection experiments using HepG2-NTCP cells or primary human hepatocytes. Conversely, cytoplasmic ELAVL1 expression has been reported to be associated with aggressive tumor phenotype [31,32]. Based on these results, this study also focused on the 13 cases with both nuclear and cytoplasmic ELAVL1 expressions, but no special clinical features were observed. Further analyses would be necessary in a larger number of HCC samples.

Pharmacological disruption of ELAVL1 would be very important for clinical application. Currently, the ELAVL1 function has multiple inhibitors, and these compounds are classified into two main categories based on their action [33]. One is drugs that inhibit the cytoplasmic transfer of ELAVL1, namely, MS-444, N-benzylcantharidinamide, Latrunculin A, and Blebbistatin. The others are small molecule compounds, including CMLD-2 and KH3, which inhibit ELAVL1 binding to target mRNAs. These drugs have been reported to inhibit the growth, metastasis, and invasion of cancer cells mainly during preclinical studies. Preclinical and clinical trials have yet to be conducted to validate their anti-HBV effects.

In conclusion, ELAVL1 was successfully demonstrated to contribute to not only HBV replication but also to HCC cell growth. It may be a potent therapeutic target for the treatment of HBV-related HCC.

## 4. Materials and Methods

### 4.1. Cell Culture

HepAD38 cells, expressing pgRNA under the control of the inducible tetracycline promoter, were cultured in D-MEM/Ham’s F-12 culture medium (Merck, Darmstadt, Germany) supplemented with 10% fetal bovine serum with or without 1 mg/mL of tetracycline to maintain the repression of HBV expression or HBV replication, respectively [34,35]. *Flag-**HBx*-overexpressed HepG2 cells were cultured in Dulbecco’s Modified Eagle Medium (Merck, Darmstadt, Germany) supplemented with 10% fetal bovine serum [36].

### 4.2. Quantitative Real-Time PCR

The total RNA of cells was extracted using RNeasy Mini Kit (Qiagen, Valencia, CA, USA). Quantitative real-time PCR was performed with an MX3000P qPCR system (Stratagene, San Diego, CA, USA) using the TB Green™ Premix Ex Taq™ II (Tli RNaseH Plus) Kit (Takara Bio Inc., Shiga, Japan) according to the manufacturer’s protocols. The sequences of primers for human *ELAVL1* and *GAPDH* were as follows: *ELAVL1* (forward 5′-CTGATGAATTCTCCCTTGTTCC-3′, reverse 5′-GGCTTGGCAAATTACACTGAA-3′) and *GAPDH* (forward 5′-CTGACTTCAACAGCGACACC-3′, reverse 5′-TAGCCAAATTCGTTGTCATACC-3′). The sequences of primers for total HBV mRNA, which amplified the region from nucleotides (nt) 1803–1894 in HBV sequences covering all HBV transcripts (3.5, 2.4, 2.1, and 0.7 kb mRNAs), and HBV 3.5 kb mRNA, which amplified the region from nt 2268 to 2390 in HBV sequences, were as follows [36]: total HBV mRNA (sense 5′-TCACCAGCACCATGCAAC-3′, antisense 5′-AAGCCACCCAAGGCACAG-3′) and HBV 3.5-kb mRNA (sense 5′-GAGTGTGGATTCGCACTCC-3′, antisense 5′-GAGGCGAGGGAGTTCTTCT-3′). The sequences of primers for *HBx* were as follows: forward 5′-TCCTTTGTTTACGTCCCGTCG-3′, reverse 5′-AGTCCGCGTAAAGAGAGGTG-3′).

### 4.3. Western Blotting

HCC cells were subjected to Western blot analyses [37]. Briefly, the lysates were separated by 8% SDS-PAGE gels (Bio-Rad Laboratories, Hercules, CA, USA) and were then transferred to a polyvinylidene difluoride membrane (Bio-Rad Laboratories). The membranes were blotted with primary antibodies against ELAVL1 (Cell Signaling Technology, Danvers, MA, USA), tubulin (Oncogene Science, Cambridge, MA, USA), and Flag (DYKDDDDK) tag (Wako Pure Chemical Industries, Osaka, Japan) and horseradish peroxidase (HRP)-conjugated secondary antibody (GE Healthcare Life Sciences, Chicago, IL, USA). The membranes were developed using Immobilon Western Chemiluminescent HRP Substrate (EMD Millipore, Burlington, MA, USA), and the signals were detected using ChemiDoc XRS Systems (Bio-Rad Laboratories). Band intensity was quantified using Image Lab 4.1 software (Bio-Rad Laboratories).

### 4.4. Measurement of HBs Antigen Levels

The HBs antigen levels in culture media were measured using CLEIA. LumiPulse Presto HBsAg-HQ was used as the assay reagent and Lumipulse L2400 as the assay device (Fujirebio, Tokyo, Japan). A positive result was determined when the measured value was 0.005 IU/mL or higher.

### 4.5. Lentiviral Production and Transduction

Lentiviral vectors (CS-H1-shRNA-EF-1a-EGFP) expressing short hairpin RNA (shRNA) targeting human *ELAVL1* (target sequence: sh-*ELAVL1*-1, 5′-GGTTTGGGCGGATCATCAACT-3′; sh-*ELAVL1*-2, 5′-GGTTTGGCTTTGTGACCATGA-3′) and *luciferase* (*Luc*) were constructed. Recombinant lentiviruses were produced as described previously [38]. The cells were transduced using a lentiviral vector in the presence of protamine sulfate (10 μg/mL; Sigma-Aldrich, St. Louis, MO, USA).

### 4.6. Cell Proliferation Assay

Cells (1 × 10^6^ cells/dish) were seeded on 6 cm dishes. Cell growth of HCC cells was assessed by trypan blue staining after 48, 96 h in culture. The relative cell viability was defined as the cell number of *ELAVL1*-knockdown HepAD38 cells divided by those of the control cells.

### 4.7. Patients and Surgical Specimens

A total of 77 pairs of tumor and adjacent non-tumor tissues were subjected to clinicopathological analyses. Written informed consent was obtained from all patients. Paraffin-embedded tumor tissues and the surrounding non-tumor tissues were examined via H&E staining and immunohistochemistry with anti-ELAVL1 antibody (Cell Signaling Technology). Based on the ELAVL1 expression in the nuclei of cells, tumor and non-tumor tissues were classified as follows: negative, partial expression (<50% of nuclei), and diffuse expression (≥50% of nuclei). HCCs with negative or partial ELAVL1 expression were classified as the ELAVL1^low^ group, whereas HCCs with diffuse ELAVL1 expression were classified as the ELAVL1^high^ group. All patients received postoperative radiological follow-up every 2–6 months. Radiological assessments were evaluated based on the response evaluation criteria in solid tumors [39]. This study was approved by the research ethics committees of the Graduate School of Medicine, Chiba University (approval number: 3300) and performed according to the Declaration of Helsinki.

### 4.8. Data Collection and Analysis from the Cancer Genome Atlas (TCGA)-Liver Hepatocellular Carcinoma (LIHC)

RNA sequencing (RNA-seq) datasets (ID: TCGA.LIHC.sampleMap/HiSeqV2) were accessed and downloaded using the UCSC Xena Browser (https://xenabrowser.net/, accessed on 1 July 2021). The RNA sequencing dataset shows the gene-level transcription estimates as in normalized log counts per million (logCPM). In total, secondary analyses were conducted on RNA-seq data from both the normal liver (*n* = 50) and primary HCC (*n* = 371).

### 4.9. Statistical Analysis

Data are expressed as mean with standard deviation (SD) or median with minimum to maximum and interquartile range (IQR). Statistical differences in the quantitative variables between groups were determined using either Student’s *t*-test, the Mann–Whitney *U* test, or Kruskal–Wallis test. The chi-squared test was used for categorical variables. The log-rank test was used to analyze survival data. The level of significance was set to *p* < 0.05. All statistical analyses were conducted using the SPSS statistical software version 27 (IBM, Chicago, IL, USA).

## Figures and Tables

**Figure 1 ijms-23-07878-f001:**
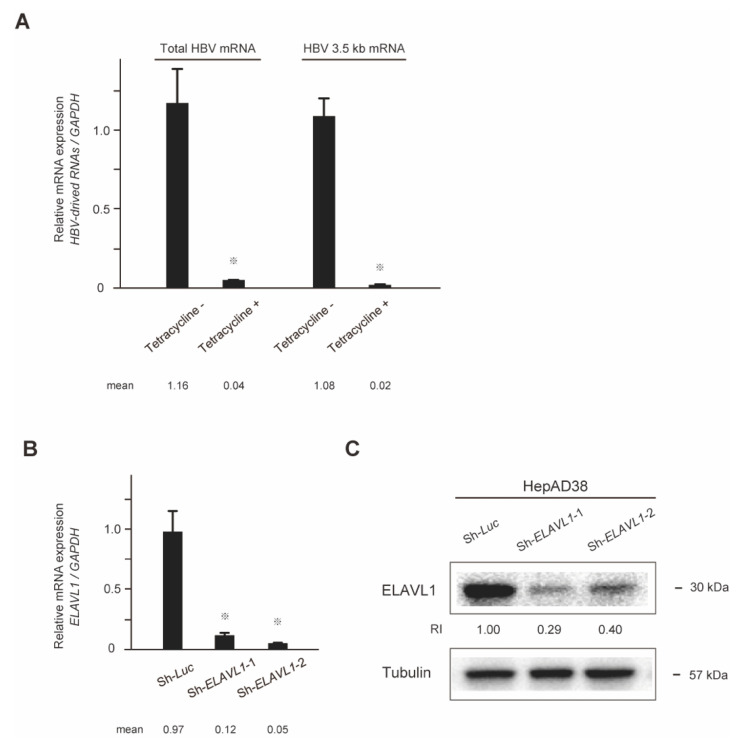
In vitro assays of HepAD38 cells. Depletion of tetracycline triggers viral replication as viral replication is under the control of a promoter that is repressed by tetracycline. (**A**) Total HBV mRNA and 3.5 kb mRNA levels of HepAD38 cells in the presence or absence of tetracycline were determined via quantitative real-time PCR analyses. Data represent the mean with SD of triplicate experiments (Student’s *t*-test, ^※^ *p* < 0.05). (**B**) *ELAVL1*-knockdown HepAD38 cells in the absence of tetracycline were subjected to quantitative real-time PCR analyses for *ELAVL1* mRNA expression. Data represent the mean with SD of triplicate experiments (Student’s *t*-test, ^※^ *p* < 0.05). (**C**) HepAD38 cells with stable knockdown of *ELAVL1* in the absence of tetracycline were subjected to Western blot analyses using anti-ELAVL1 and anti-tubulin antibody (loading control). Relative intensities (RI) were determined by normalizing band intensities to those of internal controls.

**Figure 2 ijms-23-07878-f002:**
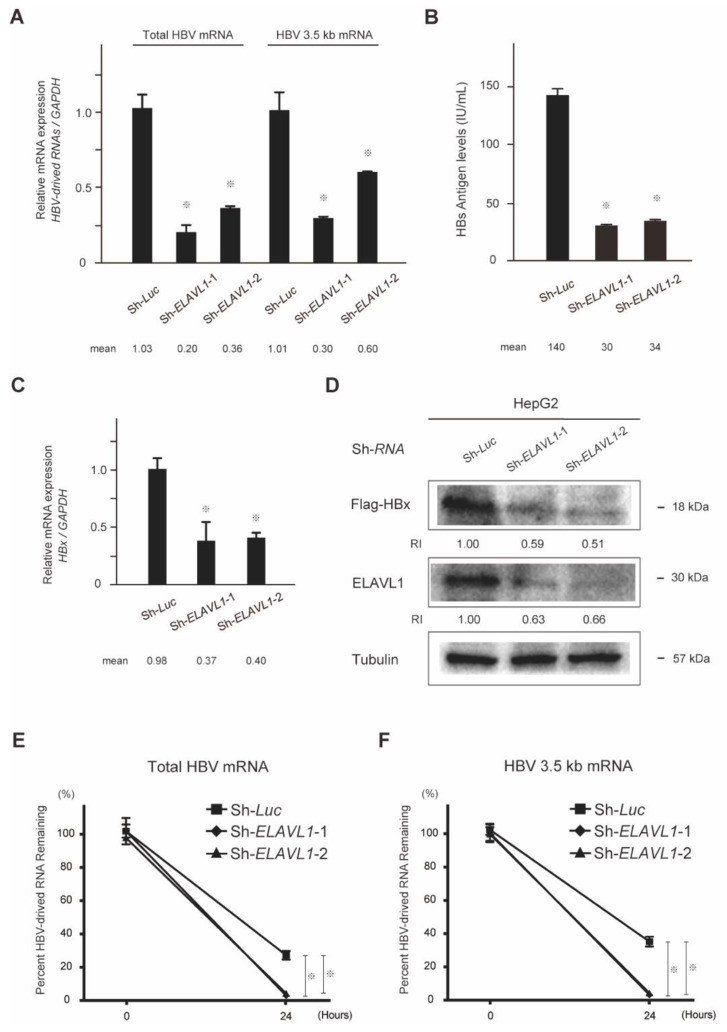
*ELAVL1*-knockdown assays in HCC cells. (**A**) Total HBV mRNA and 3.5 kb mRNA levels of *ELAVL1*-knockdown HepAD38 cells in the absence of tetracycline were determined via quantitative real-time PCR analyses. Data represent the mean with SD of triplicate experiments (Student’s *t*-test, ^※^ *p* < 0.05). (**B**) Supernatants derived from *ELAVL1*-knockdown HepAD38 cells in the absence of tetracycline were subjected to CLEIA for HBs antigen measurement. Data represent the mean with SD of triplicate experiments (Student’s *t*-test, ^※^ *p* < 0.05). (**C**) *ELAVL1*-knockdown HepG2 cells were subjected to quantitative real-time PCR analyses for *HBx* mRNA expression. Data represent the mean with SD of triplicate experiments (Student’s *t*-test, ^※^ *p* < 0.05). (**D**) *ELAVL1*-knockdown HepG2 cells that constitutively express Flag-HBx protein were subjected to Western blot analyses using anti-Flag, anti-ELAVL1, and anti-tubulin antibodies (loading control). RIs were determined by normalizing band intensities to those of internal controls. (**E**) Quantitative real-time PCR analyses of total HBV mRNA relative to GAPDH in *ELAVL1*-knockdown HepAD38 cells. RNA was harvested at 0 and 24 h post tetracycline treatment and relative levels of remaining total HBV mRNA were analyzed. Data represent the mean with SD of triplicate experiments (Student’s *t*-test, ^※^ *p* < 0.05). (**F**) Quantitative real-time PCR analyses of total 3.5 kb mRNA relative to GAPDH in *ELAVL1*-knockdown HepAD38 cells. RNA was harvested at 0 and 24 h post tetracycline treatment and relative levels of remaining 3.5 kb mRNA were analyzed. Data represent the mean with SD of triplicate experiments (Student’s *t*-test, ^※^ *p* < 0.05).

**Figure 3 ijms-23-07878-f003:**
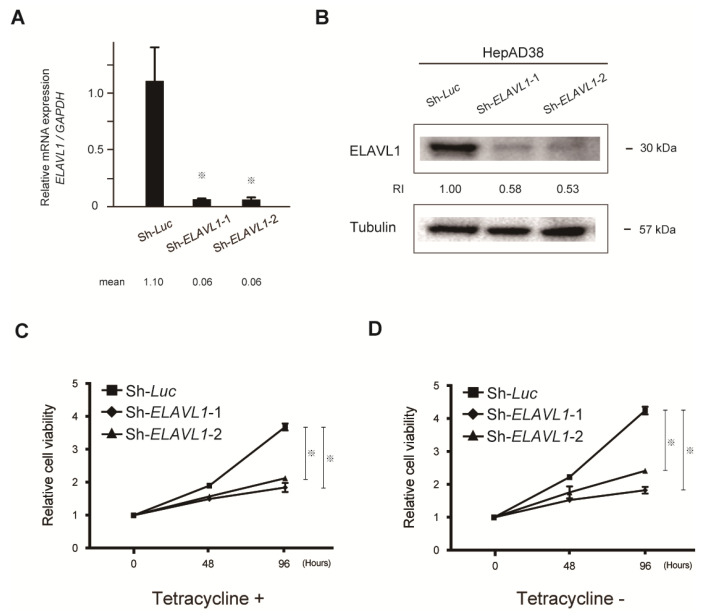
Cell growth inhibition in *ELAVL1*-knockdown HepAD38 cells. (**A**) *ELAVL1*-knockdown HepAD38 cells in the presence of tetracycline were subjected to quantitative real-time PCR analyses for *ELAVL1* mRNA expression. Data represent the mean with SD of triplicate experiments (Student’s *t*-test, ^※^ *p* < 0.05). (**B**) HepAD38 cells with stable knockdown of *ELAVL1* in the presence of tetracycline were subjected to Western blot analyses using anti-ELAVL1 and anti-tubulin antibody (loading control). RIs were determined by normalizing band intensities to those of internal controls. (**C**) Cell growth inhibition in *ELAVL1*-knockdown HepAD38 cells in the presence of tetracycline. Data represent the mean with SD of triplicate experiments (Student’s *t*-test, ^※^ *p* < 0.05). (**D**) Cell growth inhibition in *ELAVL1*-knockdown HepAD38 cells in the absence of tetracycline. Data represent the mean with SD of triplicate experiments (Student’s *t*-test, ^※^ *p* < 0.05).

**Figure 4 ijms-23-07878-f004:**
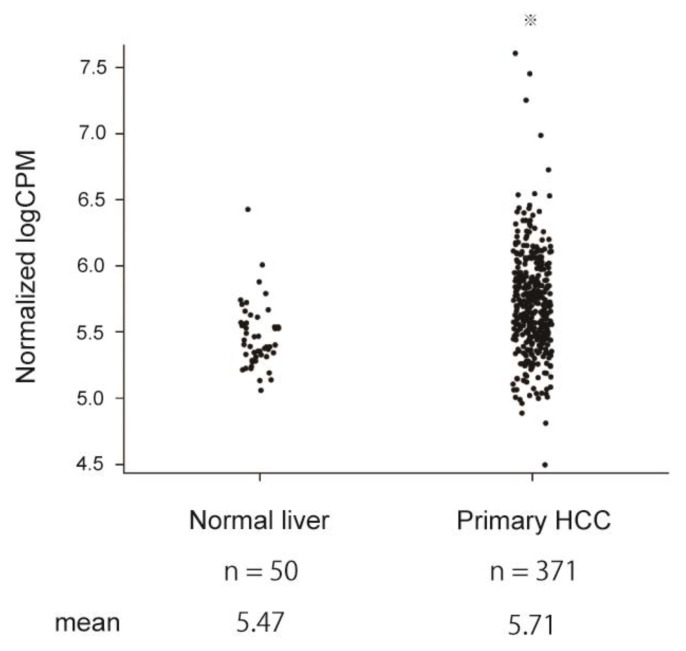
ELAVL1 expression in normal liver and primary HCC in the TCGA database. ELAVL1 mRNA expression based on the dataset obtained from TCGA-LIHC (Student’s *t*-test, ^※^ *p* < 0.05).

**Figure 5 ijms-23-07878-f005:**
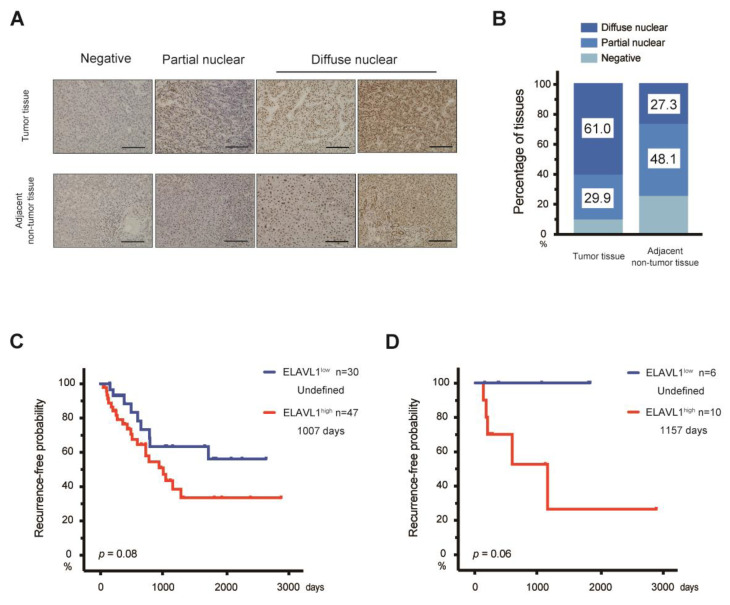
Clinicopathological analyses of ELAVL1 in HCC surgical samples. (**A**) Representative ELAVL1 immunohistochemistry analyses for tumor and adjacent non-tumor tissues. Scale bar = 100 μm. (**B**) Calculation of ELAVL1 expression patterns in tumor and adjacent non-tumor tissues. (**C**) Cumulative progression-free survival based on the ELAVL1 expression in all HCC patients (*n* = 77) (Log-rank test, *p* = 0.08). (**D**) Cumulative RFS rate based on the ELAVL1 expression in HBV-related HCC patients (*n* = 16) (Log-rank test, *p* = 0.06).

**Table 1 ijms-23-07878-t001:** Clinical features of ELAVL1^low^ and ELAVL1^high^ HCC patients.

Characteristics	ELAVL1^low^	ELAVL1^high^	*p*-Value
	(*n* = 30)	(*n* = 47)	
Age (years) (median (IQR))	67 (11)	70 (13)	0.257
Gender: male/female	23/7	37/10	0.832
Etiology: HBV/HCV/others	6/12/12	10/16/21	0.866
Fibrosis stage: CH/LC	25/5	34/13	0.266
AFP (ng/mL) (median (IQR))	12.9 (206.7)	9.2 (116.9)	0.703
BCLC stage: A/B	27/3	38/9	0.280
Edmondson–Steiner grade: I/II/III/IV	2/8/16/4	1/11/26/9	0.699

Abbreviations: HCC, hepatocellular carcinoma; HBV, hepatitis B virus, HCV, hepatitis C virus; CH, chronic hepatitis; LC, liver cirrhosis; AFP, alpha-fetoprotein; BCLC, Barcelona clinic liver cancer.

**Table 2 ijms-23-07878-t002:** Clinical features of ELAVL1^low^ and ELAVL1^high^ HBV-related HCC patients.

Characteristics	ELAVL1^low^	ELAVL1^high^	*p*-Value
	(*n* = 6)	(*n* = 10)	
Age (years ) (median (IQR))	66 (12)	62 (15)	0.713
Gender: male/female	4/2	10/0	0.051
Fibrosis stage: CH/LC	5/1	8/2	0.869
AFP (ng/mL) (median (IQR))	662.7 (1651.7)	3.9 (268.7)	0.263
BCLC stage: A/B	5/1	8/2	0.869
Edmondson–Steiner grade: I/II/III/IV	0/0/3/3	0/2/6/2	0.309

Abbreviations: HBV, hepatitis B virus, HCC, hepatocellular carcinoma; CH, chronic hepatitis; LC, liver cirrhosis; AFP, alpha-fetoprotein; BCLC, Barcelona clinic liver cancer.

## Data Availability

The data presented in this study are available from the corresponding author on reasonable request.

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
