# Peer review of "The RNA-Binding Protein ELAVL1 Regulates Hepatitis B Virus Replication and Growth of Hepatocellular Carcinoma Cells"

_ijms, 2022, doi:10.3390/ijms23147878_

Round 1

Reviewer 1 Report

In the revised manuscript, the authors made good efforts to address the concerns and comments raised by the reviewers. Nonetheless, it is still unclear as to how the RNA-binding protein ELAVL1 regulates HBV replication. It might affect viral RNA stability or replication/transcription. Based on the described roles of ELAVL1 in mRNA stability regulation in the literature, it might similarly affect viral RNA stability. However, no data are presented to support whether ELAVL1 affects viral RNA transcription or RNA stability in the revised manuscript. The authors just show ELAVL1 knockdown reduced viral RNA levels and reproduction. Any claims for effects of ELAVL1 on viral RNA stability, transcription and protein synthesis must be modified in the abstract and throughout the entire manuscript.

Minor points

1.     The descriptions for panels E and F in the legend to Figure 2 are missing

2.     It is impossible to gauge the quality and resolution of the IHC images in Figure 5A. The IHC images (also all other images including Western blots) are heavily pixelized and blurry. They are not suitable for publications. Make sure high-resolution images are used in the final publication.

Author Response

According to your suggestions, we revised the manuscript as follows:

Specific comments for the revision:

  1. In the revised manuscript, the authors made good efforts to address the concerns and comments raised by the reviewers. Nonetheless, it is still unclear as to how the RNA-binding protein ELAVL1 regulates HBV replication. It might affect viral RNA stability or replication/transcription. Based on the described roles of ELAVL1 in mRNA stability regulation in the literature, it might similarly affect viral RNA stability. However, no data are presented to support whether ELAVL1 affects viral RNA transcription or RNA stability in the revised manuscript. The authors just show ELAVL1 knockdown reduced viral RNA levels and reproduction. Any claims for effects of ELAVL1 on viral RNA stability, transcription and protein synthesis must be modified in the abstract and throughout the entire manuscript.

Reply: Thank you for your important comments. We conducted additional quantitative RT-PCR of HBV mRNAs to determine time-course RNA-stability in response to your previous suggestion. We mistakenly registered an incomplete main body, and you could not review the description based on additional experiments. We again apologize for the careless mistakes. We described these points on Line 36, Lines 130-135 and Lines 245-254 (highlighted in red) and added a new figure (Figure 2E and 2F).

Minor points

  1. The descriptions for panels E and F in the legend to Figure 2 are missing.

Reply: We apologize for careless mistakes. We added figure legends to Figs. 2E and F (highlighted in red).

  1. It is impossible to gauge the quality and resolution of the IHC images in Figure 5A. The IHC images (also all other images including Western blots) are heavily pixelized and blurry. They are not suitable for publications. Make sure high-resolution images are used in the final publication.

Reply: Thank you for your important comments. We registered high quality figures at the time of the revision and will ensure that high resolution images are also used in the final publication version in conjunction with the IJMS editorial office.

Reviewer 2 Report

Kanzaki and colleagues report about a role of Human antigen R, also known as ELAVL1, during HBV replication and growth of hepatocellular carcinoma cells. The positive of this study is large analysis of primary HCC tissues for expression of ELAVL1 and its comparison with normal liver samples. There is statistically higher expression in HCC but only a trend in the role of ELAVL1 expression in cumulative progression-free survival in both HCC patients and HBV-related HCC patients. The problem of this study is the attempt of connecting the role of ELAVL1 on HBV replication with HCC progression. The cell growth in HepAD38 cells was impaired in ELAVL1-knockdown group but it was independent of HBV replication. The ELAVL1 is known to be very promiscuous and ubiquitous RNA binding protein. There is ample evidence of the ELAVL1 role in cell signaling, fibrogenesis, inflammation and cancer development in the liver. Authors did not show substantial evidence connecting role of ELAVL1 in HBV replication and the role of ELAVL1 in HCC. Authors should show the role of ELAVL1 in real infection experiments either in HBV infection of HepG2-NTCP cells or ideally in infection of primary human hepatocytes.

Author Response

According to your suggestions, we revised the manuscript as follows:

  1. Kanzaki and colleagues report about a role of Human antigen R, also known as ELAVL1, during HBV replication and growth of hepatocellular carcinoma cells. The positive of this study is large analysis of primary HCC tissues for expression of ELAVL1 and its comparison with normal liver samples. There is statistically higher expression in HCC but only a trend in the role of ELAVL1 expression in cumulative progression-free survival in both HCC patients and HBV-related HCC patients. The problem of this study is the attempt of connecting the role of ELAVL1 on HBV replication with HCC progression. The cell growth in HepAD38 cells was impaired in ELAVL1-knockdown group but it was independent of HBV replication. The ELAVL1 is known to be very promiscuous and ubiquitous RNA binding protein. There is ample evidence of the ELAVL1 role in cell signaling, fibrogenesis, inflammation and cancer development in the liver. Authors did not show substantial evidence connecting role of ELAVL1 in HBV replication and the role of ELAVL1 in HCC. Authors should show the role of ELAVL1 in real infection experiments either in HBV infection of HepG2-NTCP cells or ideally in infection of primary human hepatocytes.

Reply: We appreciate your important comments. As pointed out, our data demonstrated that cell growth was impaired in the ELAVL1 knockdown in both HepAD38 cells treated and untreated with tetracycline. These findings indicated that ELAVL1 could regulate the cell proliferation ability of HCC cells in HBV replication-independent manners. We modified descriptions on Lines 36-39 and Lines 255-260 (highlighted in red). On the contrary, we showed that ELAVL1 affects HBx, which might be one of the reasons for the results that RFS in ELAVL1low HBV-related HCC patients tended to be better than that in ELAVL1high HBV-related HCC patients. As suggested, exploring substantial evidence connecting role of ELAVL1 in HBV replication and the role of ELAVL1 in HCC is needed to be addressed in future studies through the role of ELAVL1 in real infection experiments using HepG2-NTCP cells or primary human hepatocytes. We described these points on Lines 282-315 (highlighted in red).

Round 2

Reviewer 1 Report

Please check carefully for English spelling and grammar. 

Reviewer 2 Report

Authors substantially answered my concerns.

This manuscript is a resubmission of an earlier submission. The following is a list of the peer review reports and author responses from that submission.

Round 1

Reviewer 1 Report

The manuscript entitled “The RNA-binding protein ELAVL1 regulates hepatitis B virus replication and growth of hepatocellular carcinoma cells” by Kanzaki et al. reports potential involvement of the RNA-binding protein ELAVL1 in hepatitis B virus (HBV) replication and hepatocellular carcinoma (HCC). The data are very preliminary and much of the experimental data require validation. Some of the experiments are not done properly and lack of important controls as detailed below. The relationship between ELAVL1, HBV replication, and the growth of HCC cells is at best correlative. The manuscript sheds no lights into mechanistic insights. Overall, the conclusion is not adequately supported by the data. This manuscript is not ready for publication in its present form.

Figure 1: I assume “HBV replication +” denotes the absence of tetracycline. How would the induction of HBV replication increase ELAVL1 expression? What is the cause and what is the effect? If the authors argue that increased ELAVL1 expression promotes HBV expression, why then HBV replication in turn increases ELAVL1 expression?  The authors need to show what would be the effects of ELAVL1 overexpression on HBV replication and HCC cell growth. The effects of tetracycline on ELAVL1 expression must be confirmed with western blot as the authors did for the ELAVL1 knockdown cells (panel D). What is the reference for quantifying viral RNAs?

Figure 2: Again, the reference for quantifying viral RNAs must be specified. It is unclear how ELAVL1 depletion would downregulate the constitutively expressed Flag-HBx protein. If it is through reduced stability of the corresponding mRNA, then the authors must demonstrate the effects of ELAVL1 knockdown on the relative levels of the Flag-HBx mRNA.

Figure 3: The data do not directly show effects on cell proliferation. The authors use viability data to infer effects on cell proliferation, which is inappropriate. Panel A might just show tetracycline treatment (presumably to shut off pgRNA and there may be additional off-target effects) may cause some cells to die without affecting DNA replication and cell division. The data in panel B is impossible to interpret. The authors must include control cells and compare effects of tetracycline on control cells and cells expressing ELAVL1 shRNAs. They also must assess if tetracycline treatment results in further downregulation of ELAVL1 in knockdown cells compared to control cells at both mRNA and protein levels.

Figure 5: The IHC images at a higher resolution should be provided.

All statistical analyses should be described in corresponding figure legends.

Lines 199-200, “These results indicated that ELAVL1 may be crucial for stable RNA transcription and protein synthesis of HBV.” This statement is not supported by the reported data. ELAVL1 might stabilize viral RNA as suggested by the authors, but this may not affect transcription nor protein synthesis.

The manuscript needs careful editing to make sure each sentence is clear, and grammar is correct. For example, the sentence “HepAD38 cells that stably expressed shRNA targeting ELAVL1 or Luc were successfully achieved by lentiviral transduction” is unclear. It should be “ELAVL1 knockdown … were achieved …” Additionally, figure legends are too brief. Some details will help readers understand the figures better. For Figure 1, the authors should indicate tetracycline depletion is for inducing viral replication as viral replication is under the control of a promoter that is repressed by tetracycline.

Reviewer 2 Report

This manuscript builds on previous work to investigate mechanisms of ELAVL1 in HBV replication. ELAVL1 may interact with viral RNAs and modulate the stability or translation or act at transcription (at least as stated in the abstract. The manuscript provides some novel insights, although results are not entirely surprising. Essentially, impact of ELAVL1 depletion for HBV mRNA HBx protein levels is shown, however how ELAVL1 affects the levels remains elusive. Further information would be required to clarify this point. For instance, the authors could analyse whether ELVAL1 depletion/overexpression affects the RNA stability. At the end, the data does not show whether ELAVL1 directly actos on those RNAs under investigation or whether it relates to a secondary effect. The authors also investigate expression of ELVAL1 in HCC tumor samples, which provides a clinical angle that could be of interest. This second part shold be better described and integrated with first part as currently, those parts remain somewhat detached. 

Overall, this is a short and very concise manuscript. It is commendably careful not to over-reach with any claims, and makes sure to adequately back up their findings with appropriate data (pending further information regarding statistical testing). However, it remains unclear how ELAVL1 acts on those RNAs and further data would be required to clarify this point.

Major:

  1. Additional information how ELVAL1 affects HBV RNA levels is essential as it is not clear how current on RNA abundance levels are achieved. For instance, time-course RNA-stability assays (after transcription block) could show whether it affects RNA stability while some straightforward RNA-interaction assays could confirm RNA-protein interactions (possibly through AU-rich elements). In this regard, it is not clear why authors state that the role of ELVAL1 “in HBV transcription remains unclear” in the abstract. While ELVAL1 is an RNA-binding protein, it is more likely to have a role in post-transcriptional control rather than transcription. As such, the study remains unclear at this point and distinction between potential effects at transcription or post-transcriptional should be added for for clarification.

Minor:

  • Lines 99-100: “HBV replication caused by tetracycline depletion” could be clearer, does this mean the replication that takes place when the HCCs aren’t treated with tetracycline?
  • Lines 107-108: Add % of depletion of RNA knockdown as shown by the RT-qPCR?
  • Figure 1A and 1B: “HBV replication +” and “HBV replication -“ seem less clear than simply “Tetracycline +” and “Tetracycline -“, especially as HBV RNA is what’s being measured.
  • Figure 1D: If possible, authors should quantify Western signals to indicate knockdown efficiency at the protein level.
  • Figure 2: Data should be quantified for comparison. Essentially, quantitative analysis could be done for all other Western blots and RT-qPCRs shown in the manuscript.
  • Figure 3: As a general comment for this figure as well as others, things like how many cell counts were undertaken, and how the error bars were calculated would be beneficial.
  • Lines 128-129: What is meant by significant? P-values and statistical test used should be indicated.
  • Line 193: What is meant by tetracycline depletion?
  • Lines 220-222: Any suggestions as to why this might be?
  • Lines 282-283: Was this just according to a CLEIA Lumipulse System protocol? If so, it would be good to further expand on this.
  • Lines 321-328: I appreciate that 4.9 does address the stats to some extent. However, the test used should be added to respective section in the results section and for figure legends. Admittedly, Figure 5 is quite good about covering this!